



# A blended TROPOMI+GOSAT satellite data product for atmospheric methane using machine learning to correct retrieval biases

Nicholas Balasus[1], Daniel J. Jacob[1,2], Alba Lorente[3], Joannes D. Maasakkers[3], Robert J. Parker[4,5], Hartmut Boesch[4,5,a], Zichong Chen[1], Makoto M. Kelp[2], Hannah Nesser[1], Daniel J. Varon[1]

[1]School of Engineering and Applied Sciences, Harvard University, Cambridge, USA
[2]Department of Earth and Planetary Sciences, Harvard University, Cambridge, USA
[3]SRON Netherlands Institute for Space Research, Leiden, The Netherlands
[4]National Centre for Earth Observation, University of Leicester, Leicester, UK
[5]Earth Observation Science, School of Physics and Astronomy, University of Leicester, Leicester, UK
[a]now at: Institute of Environmental Physics (IUP), University of Bremen FB1, Bremen, Germany

*Correspondence to*: Nicholas Balasus (nicholasbalasus@g.harvard.edu)

**Abstract.** Satellite observations of dry column methane mixing ratios ($XCH_4$) from shortwave infrared (SWIR) solar backscatter radiation provide a powerful resource to quantify methane emissions in service of climate action. The TROPOMI instrument launched in October 2017 provides global daily coverage at $5.5 \times 7$ km$^2$ nadir pixel resolution but its retrievals can suffer from biases associated with SWIR surface albedo, scattering from aerosols and cirrus clouds, and across-track variability (striping). The GOSAT instrument launched in 2009 uses a retrieval method that is less subject to biases, but its data density is 200 times sparser than TROPOMI. Here we present a blended TROPOMI+GOSAT methane product obtained by training a machine learning (ML) model to predict the difference between TROPOMI and GOSAT co-located measurements, using only predictor variables included in the TROPOMI retrieval, and then applying the correction to the complete TROPOMI record from January 2018 to present. We find that the largest corrections are associated with coarse aerosol particles, high SWIR surface albedo, and across-track pixel index. Our blended product corrects a systematic difference between TROPOMI and GOSAT over water, and it features corrections exceeding 10 ppb over arid land, persistently cloudy regions, and high northern latitudes. It reduces the TROPOMI spatially variable bias over land (referenced to GOSAT data) from 14.0 to 10.7 ppb at 0.25° × 0.3125° resolution. Validation with TCCON ground-based column measurements shows reductions in variable bias compared to the original TROPOMI data from 6.0 to 5.2 ppb and in single-retrieval precision from 13.8 to 11.7 ppb. TCCON data are all in locations of SWIR surface albedo below 0.4 (where TROPOMI biases tend to be relatively low), but they confirm the dependence of TROPOMI biases on SWIR surface albedo and coarse aerosol particles, as well as the reduction of these biases in the blended product. Fine-scale inspection of the Arabian Peninsula shows that a number of hotspots in the original TROPOMI data are removed as artifacts in the blended product. The blended product also corrects striping and aerosol/cloud biases in single-orbit TROPOMI data, enabling better detection and quantification of ultra-emitters. Residual coastal biases





can be removed by applying additional filters. The ML method presented here can be applied more generally to validate and correct data from any new satellite instrument by reference to a more established instrument.

## 1 Introduction

Methane is a strong greenhouse gas, responsible for a third of the increase in global mean surface air temperature from 1750 to 2019 (Szopa et al., 2021). Its high global warming potential and short atmospheric lifetime of only 9 years (Prather et al., 2012) make it an attractive mitigation target to address near-term climate change (Nisbet et al., 2020). Monitoring progress in methane mitigation requires knowledge of worldwide emissions, but these are still highly uncertain (Saunois et al., 2020). Global satellite observations of atmospheric methane provide important top-down information to improve emission inventories

by inversion of chemical transport models (CTMs) to relate concentrations to emissions (Palmer et al., 2021). GOSAT has been in space since 2009 and provides mature and accurate retrievals, but they are relatively sparse (Parker et al., 2020). The TROPOMI instrument was launched in 2017 and provides global daily coverage, but it is more subject to biases than GOSAT because it uses a different spectral viewing window and has coarser spectral resolution (Jacob et al., 2022). Here we apply machine learning (ML) to produce a blended TROPOMI+GOSAT product that uses GOSAT to correct biases in the TROPOMI

data and enables more reliable application of these data for global inference of methane emissions.

Methane can be observed from space by nadir measurement of the spectrum of backscattered sunlight in the shortwave infrared (SWIR) spectral range. There are strong methane absorption features at 1.65 μm and 2.3 μm, enabling retrieval of the atmospheric methane column with near-unit sensitivity down to the surface under clear-sky conditions (Frankenberg et al.,

2005). Normalization of this methane column to the dry air mass yields a dry total column averaged mixing ratio of methane ($XCH_4$) as the standard retrieved quantity (Jacob et al., 2016). Retrievals can be biased when spectral structure in the surface albedo is misinterpreted as methane absorption (Jongaramrungruang et al., 2021). Poorly resolved optically thin scatterers including aerosols and cirrus clouds, as well as stray light from adjacent reflective surfaces can also bias methane retrievals (Aben et al., 2007; Butz et al., 2010; Schepers et al., 2012).


The susceptibility of methane retrievals to surface and atmospheric scattering effects depend on several factors including the spectral resolution of the instrument and the choice of SWIR band. GOSAT measures in the 1.65 μm band with 0.06 nm spectral resolution, enabling accurate retrieval of methane using the proxy approach that takes advantage of $CO_2$ absorption in that same band (Parker et al., 2011). The $CO_2$ proxy approach multiplies the $XCH_4/XCO_2$ ratio retrieved without consideration

of atmospheric scattering by a local $XCO_2$ value from a CTM calibrated with observations. This takes advantage of the much smaller variability of $XCO_2$ than $XCH_4$ and largely cancels surface and atmospheric artifacts. A limitation of the proxy approach is the assumption of accurate prior $XCO_2$, which can introduce biases when $CO_2$ and methane are co-emitted from a flare, for example. The proxy approach has demonstrated accuracy (Buchwitz et al., 2015) and the GOSAT retrievals are



mature. The main limitation of GOSAT is its sparsity of observations, which are taken in 10.5 km diameter pixels spaced about 270 km apart with a return time of 3 days (Kuze et al., 2016). The GOSAT data have been used extensively for inversions of methane emissions on global and continental scales (Turner et al., 2015; Maasakkers et al., 2019; Janardanan et al., 2020; Western et al., 2021; Maasakkers et al., 2021; Qu et al., 2021; Worden et al., 2022; Feng et al., 2022).

TROPOMI provides global daily coverage in continuous $5.5 \times 7$ km$^2$ (nadir) pixels, increasing the data density relative to GOSAT by more than two orders of magnitude. It measures in the 2.3 μm band, where the $CO_2$ proxy approach is not possible, with a spectral resolution of 0.25 nm. Retrieval of $XCH_4$ by TROPOMI employs a full-physics approach in which surface albedo and atmospheric scattering properties are retrieved together with $XCH_4$, utilizing additional information from the near-infrared (NIR) band of TROPOMI (Butz et al., 2012). Aliasing between these parameters in the retrieval can produce artifacts that bias the inference of methane emissions (Barré et al., 2021; Qu et al., 2021; Jacob et al., 2022). Recent improvements to

the operational retrieval produced by the Netherlands Institute for Space Research (SRON) have reduced some of these biases (Lorente et al., 2021; 2022b). An independent TROPOMI retrieval by the University of Bremen (Schneising et al., 2019; 2023) applied a ML correction to a methane climatology to remove retrieval biases, but this may bias the product if the correction to climatology is not appropriate.

Our blended TROPOMI+GOSAT methane product aims to eliminate biases from the TROPOMI data by using co-located GOSAT methane retrievals from four years of observations (2018-2021) to train a ML model for predicting the TROPOMI-GOSAT $XCH_4$ difference, relying only on predictor variables included in the TROPOMI methane product. This allows us to apply the TROPOMI-GOSAT correction to all TROPOMI data (2018-present) to form the blended product. The ML model also identifies the main sources of bias in the TROPOMI data to guide further improvements in the retrieval. The methods

presented here are not specific to TROPOMI and GOSAT and could be applied to any other satellite instrument pairs.

## 2 Construction of the blended TROPOMI+GOSAT product

Table 1 summarizes the GOSAT and TROPOMI data used in the construction of our blended TROPOMI+GOSAT product including the GOSAT v9.0 proxy retrieval from Parker et al. (2020) (quality flag = 0) and the TROPOMI SRON v19 retrieval from Lorente et al. (2022b) (quality assurance value = 1, albedo bias-correction applied). The TROPOMI SRON v19 retrieval

is consistent with v02.04.00 of the operational product. It is standard practice to evaluate satellite methane products with ground-based $XCH_4$ observations from the Total Carbon Column Observing Network (TCCON) (Wunch et al. 2011). We do so here for the GOSAT and TROPOMI retrievals using the 24 TCCON sites available in the GGG2020 version of the data during 2018-2021, adjusting all retrievals to common vertical profiles and averaging kernel sensitivities as described in Appendix A. Details of the evaluation with TCCON data are given in Appendix B and results are given in Table 1. All of the

TCCON sites are over land and most are at northern mid-latitudes. We calibrate GOSAT to have a global mean bias of 0 ppb



relative to GGG2020 TCCON data, subtracting 8.9 ppb from all retrievals. This follows Parker et al. (2020) but updates the TCCON data version that is calibrated against from GGG2014 to GGG2020. TROPOMI has a global mean bias of 4.6 ppb relative to TCCON. The standard deviation of the satellite-TCCON difference for individual retrievals gives a measure of retrieval precision and is 15.0 ppb for GOSAT and 13.8 ppb for TROPOMI. Most critical for inversions is the spatially variable

bias, which reflects artifact data features that inversions could interpret as emissions. Variable bias is commonly diagnosed with TCCON data as the standard deviation of the temporally averaged satellite-TCCON differences for individual stations. Table 1 gives variable biases relative to TCCON of 5.2 ppb for GOSAT and 6.0 ppb for TROPOMI, which are lower than the 10 ppb threshold defined by Buchwitz et al. (2015) for successful regional inversions. However, the spatial coverage of TCCON stations for estimating this variable bias is very limited as the stations are mainly located in regions of moderate SWIR

surface albedo (Figure B1) where retrieval biases tend to be low (Lorente et al., 2021). A full global diagnostic of variable bias based on TROPOMI-GOSAT differences shows much larger values depending on region (Qu et al., 2021; Jacob et al., 2022).

We compute TROPOMI-GOSAT differences Δ(TROPOMI-GOSAT) for all co-located individual retrievals from 1 January 2018 to 31 December 2021 (four years of data), adjusting all retrievals to common prior vertical profiles and averaging

kernel sensitivities as described in Appendix A. Co-location is defined by pixel centers ≤ 5 km apart and retrieval times ≤ 1 hour apart, resulting in 159859 pairs for the four years including 152480 pairs over land and 7379 over water. Data are much sparser over the oceans and limited to lower latitudes because of requirement for specular reflectance in the glint retrieval. The standard deviation of the difference for individual data pairs is 16.7 ppb.

**Table 1.** TROPOMI and GOSAT data used for the blended TROPOMI+GOSAT product.

| | GOSAT | TROPOMI |
|---|---|---|
| **Retrieval Version** | UoL v9.0 [a] | SRON v19 [b] |
| **Local Overpass Time** | 13:00 | 13:30 |
| **Pixel Size** | 10.5 km diameter | $5.5 \times 7$ km$^2$ [c] |
| **Pixel Separation** | 260-280 km | none |
| **Coverage** | global | global |
| **Return Time** | 3 days | 1 day |
| **Retrieval Type** | $CO_2$ proxy at 1.65 μm | full physics at 2.3 μm |
| **Number of Retrievals per Day** [d] | 1454 | 308279 |
| **Mean Bias** [e] | 0.0 ppb [f] | 4.6 ppb |
| **Variable Bias** [e] | 5.2 ppb | 6.0 ppb |
| **Single-Retrieval Precision** [e] | 15.0 ppb | 13.8 ppb |



[a] Parker et al. (2020). Only observations with quality flag = 0 are used.

[b] Lorente et al. (2022b). Only observations with quality assurance value = 1 are used. Albedo bias-corrected data ("xch4_corrected") is used.

[c] At nadir; $7 \times 7$ km$^2$ before 6 August 2019.

[d] Average for 2018-2021.

[e] Based on differences with TCCON data (version GGG2020) as derived in this work. See Appendix B for details. Variable bias is the spatial standard deviation of the temporally averaged differences for individual TCCON stations. Retrieval precision is the standard deviation of the differences for individual retrievals.

[f] In this work, we calibrate the UoL v9.0 product to zero global mean bias relative to TCCON (version GGG2020), subtracting 8.9 ppb from all retrievals.

Figure 1 shows the average difference Δ(TROPOMI-GOSAT) for 2018-2021, plotted on a $2° \times 2.5°$ grid for visualization purposes. The global mean bias of TROPOMI relative to GOSAT taken as reference is 1.9 ppb over land and 12.0 ppb over water. Despite the low global mean bias over land, there are large areas with over 20 ppb differences including over bright surfaces (North Africa), persistently cloudy areas (Amazon, Congo, Southeast Asia), and snow-covered surfaces (high northern latitudes).

We quantify a spatially variable bias for TROPOMI relative to GOSAT using the same definition as used for TCCON (spatial standard deviation of the temporally averaged differences). We do this for spatial resolutions of $0.25° \times 0.3125°$ and $2° \times 2.5°$, typical of regional and global inversions respectively. We separate land and water because users conducting inversions may choose not to use the glint data over water. At $0.25° \times 0.3125°$ resolution we find variable biases of 13.2 ppb over land and 15.0 ppb over water, while at $2° \times 2.5°$ resolution we find variable biases of 11.4 ppb over land and 12.4 ppb over water. These variable biases imply that inversions using GOSAT or TROPOMI would produce significantly different results. Variable biases do not decrease much in going from $0.25° \times 0.3125°$ to $2° \times 2.5°$, suggesting that most of the biases are large-scale regional features.

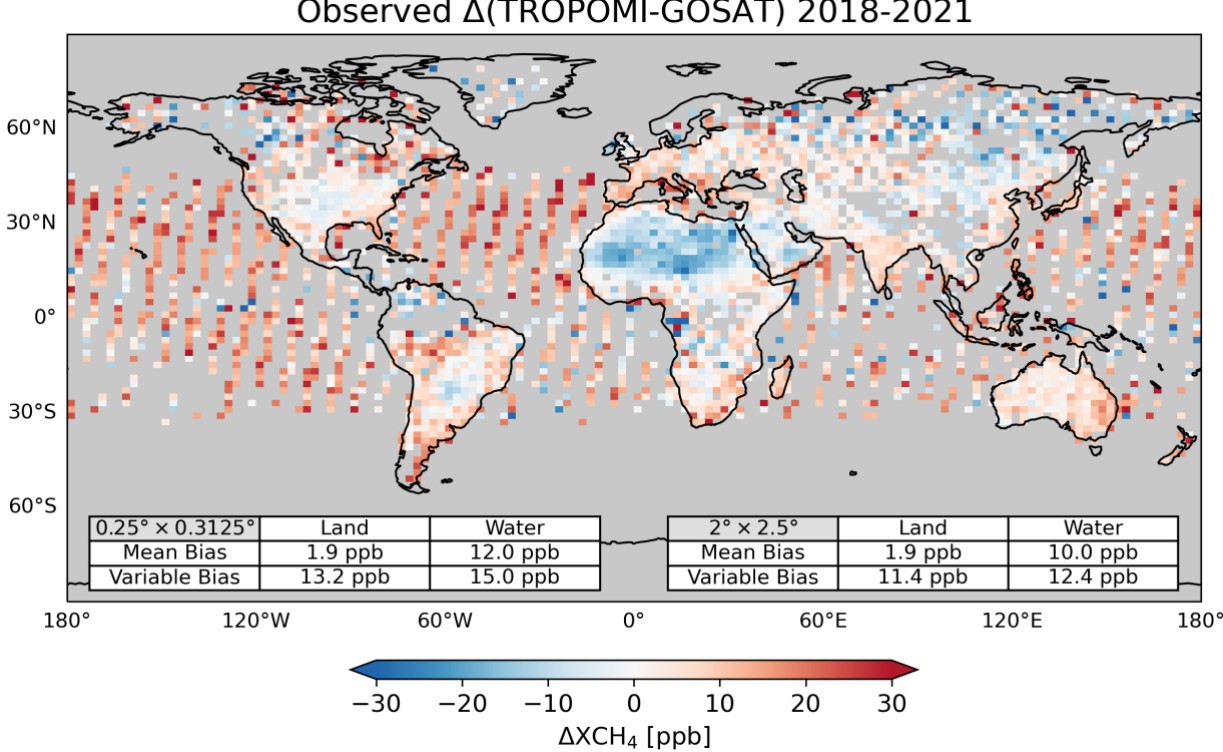

**Figure 1: Average difference Δ(TROPOMI-GOSAT) between co-located GOSAT and TROPOMI observations for 2018-2021, plotted on a 2° × 2.5° grid for visibility. Co-location criteria are observation times within 1 hour and pixel centers within 5 km. The GOSAT and TROPOMI observations have been adjusted to common prior estimates and averaging kernel sensitivities to enable meaningful computation of differences (Appendix A). Mean bias and variable bias of TROPOMI relative to GOSAT are shown inset separately over land and water (data over water are from the glint product). Mean bias and variable bias are calculated respectively as the spatial average and standard deviation of the temporally averaged Δ(TROPOMI-GOSAT) data on the specified grid (0.25° × 0.3125° or 2° × 2.5°).**

We use the co-located GOSAT and TROPOMI data for 2018-2021 to develop a predictive ML model for Δ(TROPOMI-GOSAT) that can be applied to correct the TROPOMI data with reference to the GOSAT data. The model uses the 30 predictor variables listed in Table 2, which are all TROPOMI retrieval parameters included with the individual XCH$_4$ observations, so that the correction can then be applied to the full TROPOMI dataset as a function of those parameters. We split the 159859 co-located data pairs into two sets. The pairs for 2018-2020 are used to train the predictive model (training dataset). The training minimizes a loss function of mean squared error that describes the difference between predicted and true Δ(TROPOMI-GOSAT) values. The pairs for 2021 are used for independent evaluation of the predictive model (test dataset).



We considered three candidate ML methods (Random Forest, LightGBM, and XGBoost) that rely on ensembles of decision trees (Kingsford and Salzberg, 2008). Random Forest grows an ensemble of decision trees using a bootstrapped sample of training data and subset of features for each decision tree. The averaged predictions from the forest of trees form the model prediction (Breiman, 2001). LightGBM and XGBoost are different implementations of gradient-boosted decision tree algorithms in which decision trees are grown sequentially with each iteration predicting the residual between the observation

and the sum of all previous decision trees (Ke et al., 2017; Chen and Guestrin, 2016). It is not necessary to normalize the predictor variables from Table 2 for any of these methods. To choose the best predictive model, we used their implementation in Microsoft's Fast and Lightweight AutoML Library (FLAML) (Wang et al., 2021). FLAML is designed to select the ML model (both method and hyperparameters) that would perform best on the test data. To keep the test data independent, FLAML evaluates models with 10-fold cross validation on the training data (with the 10 folds determined by dividing the data

sequentially). LightGBM performed the best, XGBoost was a close second, and Random Forest did the worst by a significant margin. We perform a Z-score transform on model predictions to account for systematic bias in ensemble-tree based approaches at extreme values as was done by Ouyang et al. (2023) and is explained by Belitz and Stackelberg (2021).

**Table 2.** TROPOMI retrieval parameters used to predict $\Delta$(TROPOMI-GOSAT). [a]

| Predictor Variable | Units |
| --- | --- |
| 1. Solar Zenith Angle | degree |
| 2. Relative Azimuth Angle | degree |
| 3. Across-Track Pixel Index [b] | -- |
| 4. Land Flag [c] | -- |
| 5. Surface Altitude | m |
| 6. Surface Roughness | m |
| 7. U10 Wind Speed [d] | $m\ s^{-1}$ |
| 8. V10 Wind Speed [d] | $m\ s^{-1}$ |
| 9. $XCH_4$ *a priori* | ppb |
| 10. Cirrus Reflectance [e] | -- |
| 11. $XCH_4$ Precision [f] | ppb |
| 12. Fluorescence [g] | $photons\ s^{-1}\ cm^{-2}\ nm^{-1}\ sr^{-1}$ |
| 13/14. CO Column and Precision | $molecules\ cm^{-2}$ |
| 15/16. $H_2O$ Column and Precision | $molecules\ cm^{-2}$ |
| 17/18. Aerosol Size Distribution Parameter and Precision [h] | -- |
| 19/20. Aerosol Height and Precision [i] | m |
| 21/22. Aerosol Column and Precision | $particles\ cm^{-2}$ |





| | |
|---|---|
| 23/24. SWIR Surface Albedo and Precision [j] | -- |
| 25/26. NIR Surface Albedo and Precision [k] | -- |
| 27. SWIR Aerosol Optical Thickness | -- |
| 28. NIR Aerosol Optical Thickness | -- |
| 29. SWIR Chi-Squared [l] | -- |
| 30. NIR Chi-Squared [l] | -- |


[a] All 30 parameters in this Table are provided together with XCH$_4$ as part of the individual SRON v19 TROPOMI methane retrievals. They are used in the LightGBM machine learning (ML) algorithm to predict Δ(TROPOMI-GOSAT) for individual TROPOMI retrievals.

[b] The retrieval also provides satellite viewing angle but this is redundant with the across-track pixel index.

[c] Surface classification for land is from the 1-km resolution Global Land Cover Characteristics Data Base Version 2.0 (USGS, 2018) and for water from the 250-m resolution data from Carroll et al. (2009) as explained by Apituley et al. (2022). This parameter has four possible values: 0 = land, 1 = water, 2 = mostly land (with some water), 3 = mostly water (with some land).

[d] Zonal and meridional wind speeds at 10-meter altitude.

[e] From the Visible Infrared Imaging Radiometer Suite (VIIRS).

[f] Precision as given in the TROPOMI retrieval product only includes the effect of noise in the measured radiance and is much smaller than the retrieval precision given in Table 1 from validation with TCCON data (Lorente et al., 2021).

[g] Fluorescence emission at 755 nm.

[h] Negative power law exponent (α) for the aerosol size distribution represented as $n(r) \sim r^{-\alpha}$ where $n$ is the number size

distribution function and $r$ is particle radius (Hasekamp et al., 2022). Larger values of α correspond to a finer aerosol.

[i] Central height of Gaussian aerosol altitude distribution (Hasekamp et al., 2022).

[j] Shortwave infrared (SWIR) at 2305-2385 nm.

[k] Near-infrared (NIR) at 757-774 nm.

[l] Quantifies goodness of fit for the retrieval in the SWIR or NIR spectral band.


     We applied the SHapley Additive exPlanations (SHAP) approach to determine the contributions of the individual variables in Table 2 to the prediction of Δ(TROPOMI-GOSAT). SHAP analysis partitions individual model predictions to the different predictor variables, giving each a SHAP value (in unit of ppb) that add up to the deviation of the model prediction from the average prediction across a given dataset. We use the TreeExplainer method for our SHAP analysis (Lundberg et al., 2020).

The SHAP values for the predictor variables can be used to understand individual predictions, or they can be aggregated across a larger set of data. They do not fully resolve correlation across predictor variables, which can complicate interpretability (Aas et al., 2021; Silva et al., 2022).





Figure 2 ranks the predictor variables of Table 2 by their average absolute SHAP values across the training dataset. The
most important predictors of Δ(TROPOMI-GOSAT) are the aerosol size distribution parameter (given by the negative power
law exponent), the SWIR surface albedo, and the across-track pixel index. SWIR aerosol optical thickness is strongly correlated
with the aerosol size distribution parameter ($R^2 = 0.69$), but the SHAP analysis does not resolve this correlation. Although one
might expect arid surfaces to generate large dust particles, we find that SWIR surface albedo is not correlated with the aerosol
size distribution parameter ($R^2 = 0.02$). The importance of the across-track pixel index reflects the striping patterns present in
TROPOMI retrievals (Borsdorff et al., 2018).

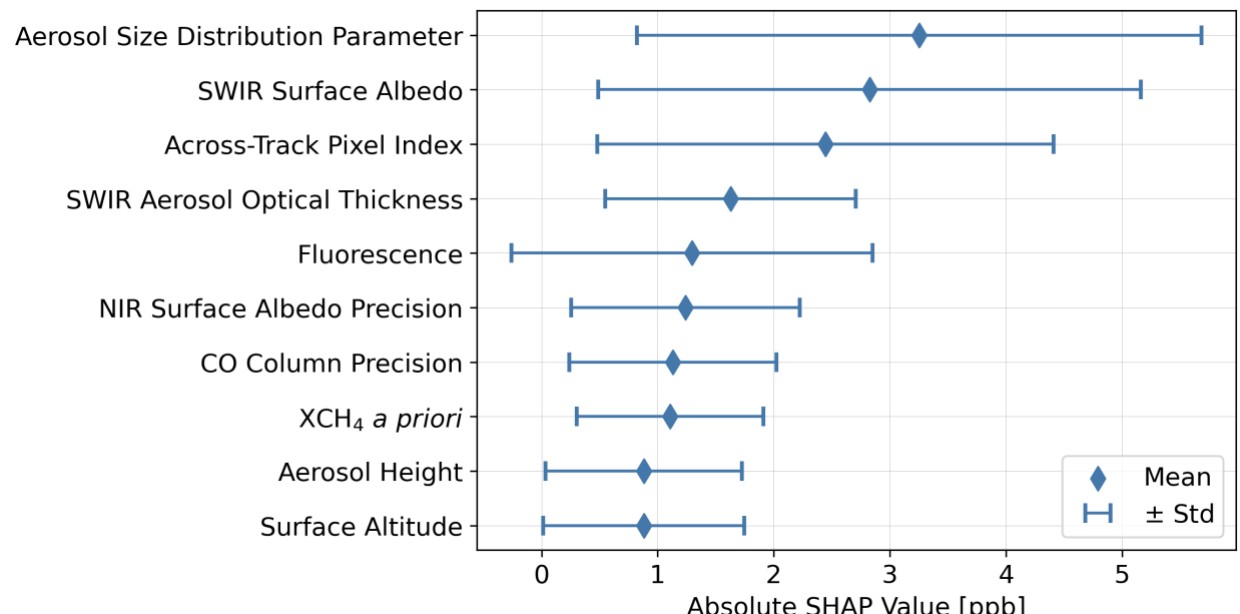

**Figure 2: Predictors of Δ(TROPOMI-GOSAT) ranked in order of importance. The Figure shows the top ten
predictor variables for the ML model of Δ(TROPOMI-GOSAT) among all predictor variables included in the**
**TROPOMI retrieval dataset (Table 2). The contributions of individual variables are defined by their mean absolute
SHAP values in unit of ppb and are shown here as global mean absolute values and standard deviations for the
training data of 2018-2020.**

Figure 3 further examines the SHAP values for the aerosol size distribution parameter (with smaller values indicating
larger particles) and SWIR surface albedo. TROPOMI data appear to be biased low with respect to GOSAT when particles are
large, which is a recognized source of error for full-physics retrievals (Butz et al., 2010; Schepers et al., 2012). Despite already
undergoing a bias correction with respect to albedo (Lorente et al., 2021), TROPOMI data are biased low relative to GOSAT
at high SWIR surface albedo.



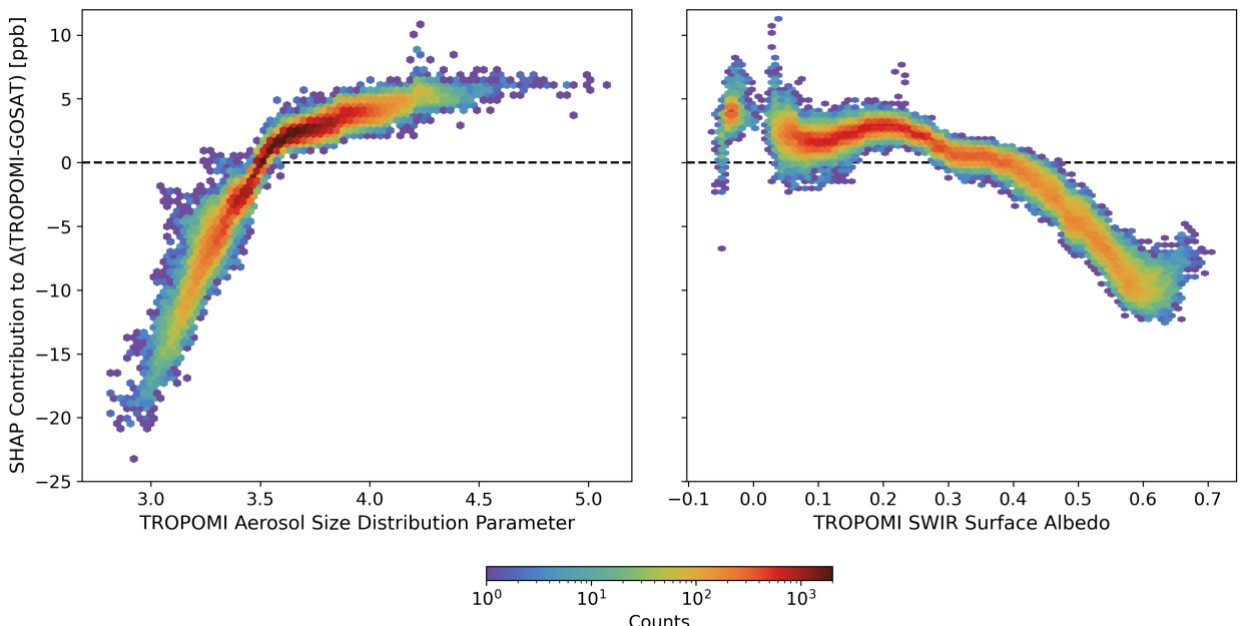

**Figure 3: Contributions to Δ(TROPOMI-GOSAT) from the two TROPOMI retrieval parameters of most importance: aerosol size distribution parameter and SWIR surface albedo. The aerosol size distribution parameter is the negative exponent of the assumed power law aerosol size distribution in the TROPOMI retrieval (Table 2) and decreases as the contribution from large particles increases. The SWIR surface albedo is for the 2305-2385 nm wavelength range. Negative values are for water scenes where the SWIR surface albedo is calculated differently in the retrieval (Lorente et al., 2022a). The figure shows the SHAP-inferred contributions of the two parameters to the predicted Δ(TROPOMI-GOSAT) values for individual data pairs (counts) in the 2018-2020 training dataset of the ML model.**

## 3 Evaluation of the blended TROPOMI+GOSAT product

Figure 4 shows the ability of the ML model to predict Δ(TROPOMI-GOSAT) for the 2021 testing data that the model was not trained on. The correction is overall successful, with a coefficient of determination ($R^2$) of 0.49 and a root-mean-square-error (RMSE) of 12.7 ppb. Random noise necessarily limits the quality of the fit for individual pairs. The RMSE is smaller than would be expected from the precision of Δ(TROPOMI-GOSAT) derived by adding the precisions of the individual TROPOMI and GOSAT retrievals relative to TCCON in quadrature (20.4 ppb; Table 1). This implies that the TROPOMI and GOSAT retrieval precisions derived from TCCON are not fully random but are partly predictable on the basis of the TROPOMI retrieval parameters.





Despite the bias correction applied by the Z-score transform from Belitz and Stackelberg (2021), we see from Figure 4 a tendency for the ML model to underestimate the high tail of the observed distribution and overestimate the low tail. This is a recognized problem in ML algorithms that aim to provide a good model of the mean (Zhang and Lu, 2012).


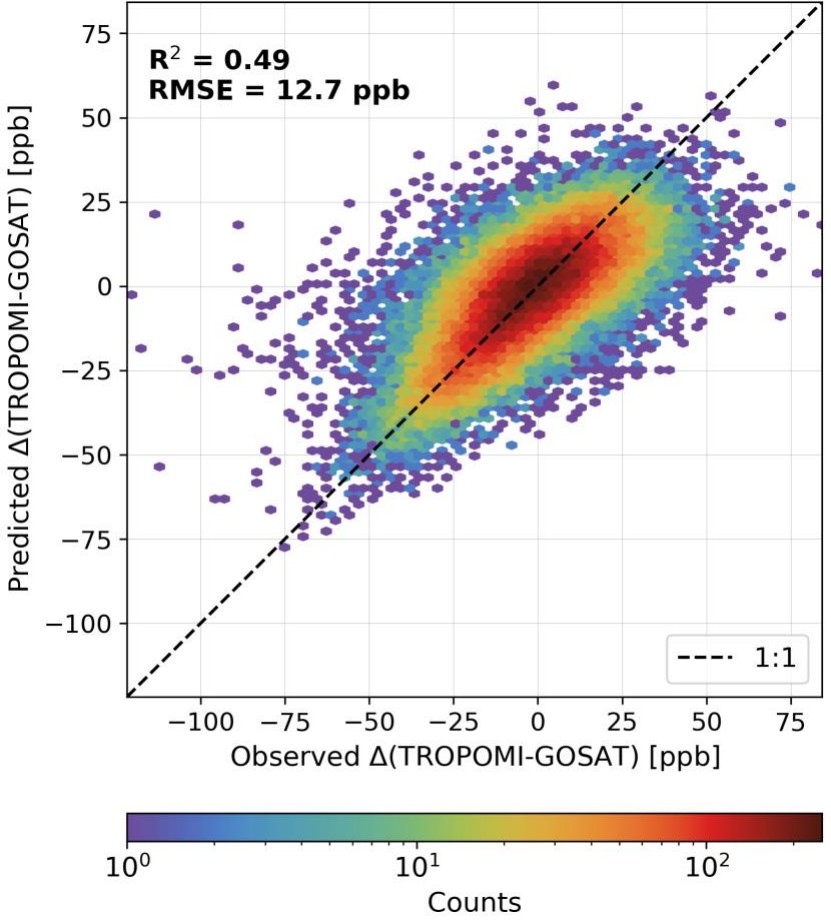

**Figure 4: Ability of the ML model to predict Δ(TROPOMI-GOSAT) on the test data from 2021. Coefficient of determination ($R^2$) and root-mean-square error (RMSE) are shown inset.**

250 Figure 5 displays the observed global distribution of Δ(TROPOMI-GOSAT) for the 2021 test data and the residual distribution (observed-predicted) after correction with the ML model. As the model was not trained on these data, the data can be used for an independent global evaluation of the reduction in the mean and variable bias of TROPOMI relative to GOSAT. Much of the original regional structure in the TROPOMI bias has disappeared or is greatly reduced. The variable bias over land decreases from 14.0 to 10.7 ppb at 0.25° × 0.3125° resolution, and from 12.8 to 9.6 ppb at 2° × 2.5° resolution. The mean





bias is reduced over water, going from 11.7 to -2.7 ppb at 0.25° × 0.3125° resolution and from 10.1 to -3.3 ppb at 2° × 2.5° resolution.

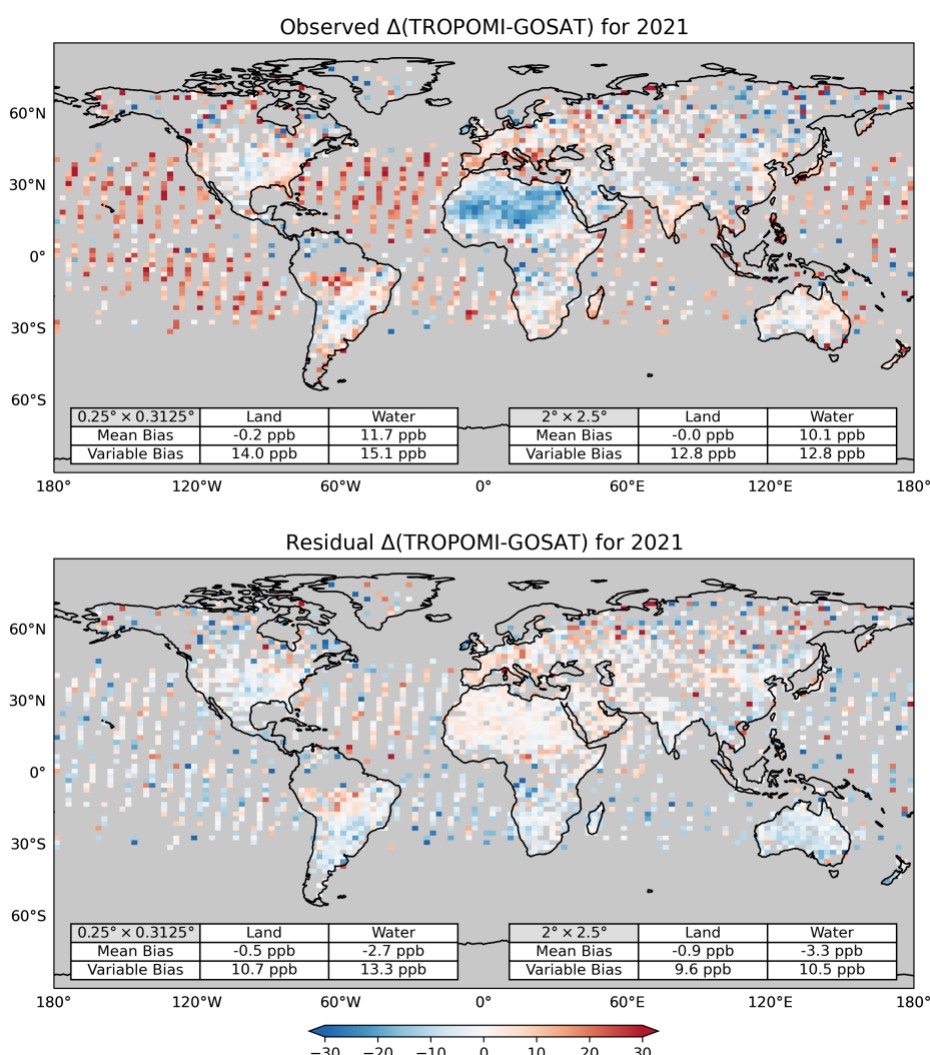

**Figure 5: TROPOMI-GOSAT XCH₄ differences (Δ(TROPOMI-GOSAT)) for co-located data in 2021, plotted on a 2°**
**× 2.5° grid for visibility. Values are annual means. The 2021 observations (top panel) are used as test data for the ML model trained to predict Δ(TROPOMI-GOSAT) from 2018-2020 data. The bottom panel shows the residual Δ(TROPOMI-GOSAT) after removing the predicted values from the observations. The bottom panel is equivalent to Δ(Blended-GOSAT). Mean bias and variable bias are calculated as described in Figure 1.**





After forming the full blended TROPOMI+GOSAT product for the 2018-2021 period (described below), we perform an independent evaluation with the TCCON data for that period covering 24 sites (Figure 6). The evaluation procedure is described in Appendix B. This allows us to compare to the evaluations of the original TROPOMI and GOSAT retrievals with the same TCCON data. We find that the retrieval precision is improved from 13.8 ppb in the TROPOMI data to 11.7 ppb in the blended TROPOMI+GOSAT product, both surpassing GOSAT's precision of 15.0 ppb. The variable bias is reduced from

6.0 ppb in the TROPOMI data to 5.2 ppb for the blended TROPOMI+GOSAT product, aligning with GOSAT's variable bias of 5.2 ppb. The mean bias decreases from 4.6 ppb to -3.3 ppb. Individual station comparisons are in Table B2. All stations except for one see a reduction in the standard deviation of Δ(satellite-TCCON).

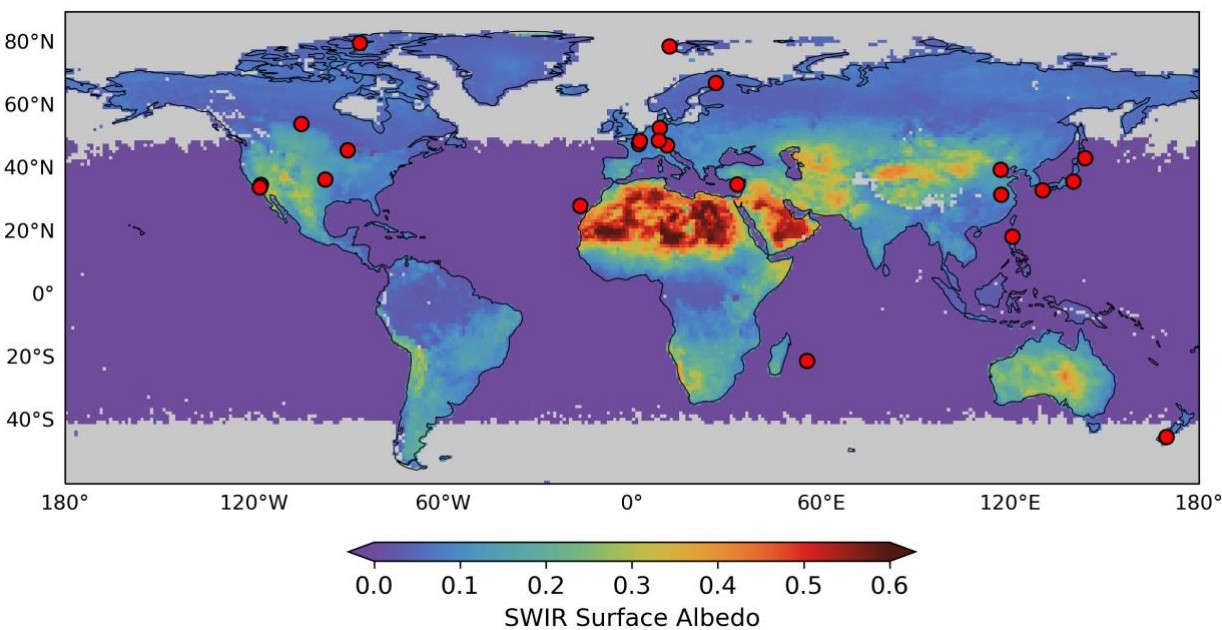

**Figure 6: TCCON stations with 2018-2021 data mapped on the mean TROPOMI SWIR surface albedo for 2021. The TROPOMI data are annual means and plotted on a 1° × 1° grid with data south of 60°S cropped for visualization purposes. Grey areas have no TROPOMI data. Site locations are listed in Table B1.**

        The blended TROPOMI+GOSAT product shows only a modest improvement in error statistics at TCCON sites, but this

is because these sites are all in locations of SWIR surface albedo lower than 0.4. As shown in Figure 3, the largest TROPOMI biases are for SWIR surface albedos higher than 0.4 (17% of all TROPOMI data). Beyond the simple evaluation, the TCCON data allow us to test our previously derived relationships of TROPOMI retrieval biases to retrieval parameters, including SWIR surface albedo and aerosol size parameter found to be most important (Figures 2 and 3). We show in Figure 7 the satellite-TCCON differences for individual co-located retrievals as a function of these two parameters. Although the TCCON data



cover only a limited SWIR surface albedo range, the TROPOMI SRON v19 retrieval shows an albedo-dependent bias with a slope of -29.1 ppb that is reduced to -10.0 ppb in the blended product. This magnitude of reduction in slope is not sensitive to removal of the water scenes with negative albedos (slopes change to -31.1 ppb and -12.2 ppb, respectively). The TROPOMI SRON v19 retrieval also shows a bias dependent on the aerosol size parameter with a slope of +21.0 ppb that is reduced to -10.2 ppb in the blended product.


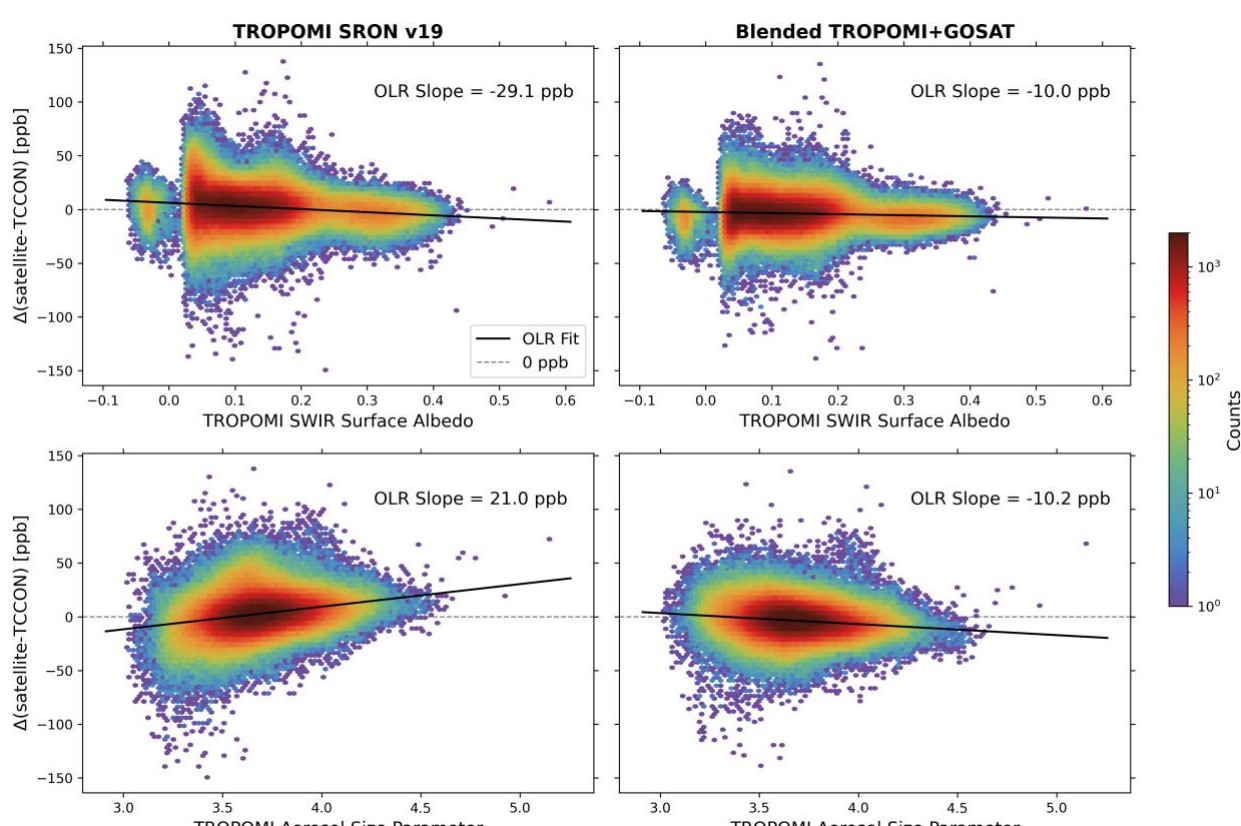

**Figure 7: Application of TCCON data to evaluate satellite XCH$_4$ retrieval biases in relation to retrieval parameters. The figure shows satellite-TCCON differences for individual co-located retrievals as a function of the TROPOMI retrieval parameters found to be the most important causes of retrieval bias (Figure 2): SWIR surface albedo (top) and**
**aerosol size parameter (bottom). Results for the TROPOMI SRON v19 retrieval (left) and the blended TROPOMI+GOSAT retrieval (right) are compared. The bias dependences on the retrieval parameters are quantified as the slopes of the ordinary linear regression (OLR) lines. Results are for the 2018-2021 period at the 24 TCCON sites of Figure 6 (excluded sites are explained in Appendix B). Eight outlier points that exceed a bias of magnitude 150 ppb are excluded from the plots. Negative values for SWIR surface albedo are for water scenes where the SWIR surface**
**albedo is calculated differently in the retrieval (Lorente et al., 2022a).**



Table 3 summarizes the error statistics of the blended TROPOMI+GOSAT product. Mean biases are low. Variable bias relative to TCCON is low (5.2 ppb), but this reflects the favorable locations of the TCCON stations as discussed above. Variable bias relative to GOSAT is about 10 ppb. This is sufficiently low that inversions of the blended TROPOMI+GOSAT

product to infer methane emissions should be consistent with inversions of GOSAT data (Buchwitz et al., 2015). The blended TROPOMI+GOSAT product benefits from the TROPOMI coverage to produce a data density ~200× higher than GOSAT.

**Table 3.** Summary of error statistics for the blended TROPOMI+GOSAT product.

| Reference Dataset | Mean Bias (ppb) | Variable Bias (ppb) |
|---|---|---|
| GOSAT (2021, 0.25° × 0.3125°, Land) | -0.5 | 10.7 |
| GOSAT (2021, 0.25° × 0.3125°, Water) | -2.7 | 13.3 |
| GOSAT (2021, 2° × 2.5°, Land) | -0.9 | 9.6 |
| GOSAT (2021, 2° × 2.5°, Water) | -3.3 | 10.5 |
| TCCON (2018-2021, GGG2020) | -3.3 | 5.2 |

**4 Overview of the blended TROPOMI+GOSAT product**

We produced a blended TROPOMI+GOSAT product for the 2018-2021 period by applying the predictive model for Δ(TROPOMI-GOSAT) to the SRON v19 TROPOMI data product. The correction is implemented as Δ(TROPOMI-GOSAT) subtracted from the TROPOMI data. The blended product contains all successful TROPOMI retrievals from January 2018 to present. Figure 8 shows the global distribution of the blended product for 2021 and the corrections to the TROPOMI retrieval.

We see a systematic downward correction over the oceans (-12.7 ± 9.6 ppb) except in persistently cloudy regions near the equator. Over land, the correction averages -5.4 ± 9.8 ppb. It is highest over bright arid surfaces, which are known to be difficult for TROPOMI retrievals (Lorente et al., 2021; Schneising et al., 2019). We also see large corrections at high northern latitudes that are seasonally driven (see below), and over tropical wetlands (Amazon, central Africa) where TROPOMI data are particularly sparse (Qu et al., 2021). Data south of 60°S (where the correction averages -8.9 ± 9.5 ppb) are excluded from these

statistics and visualizations because of a lack of GOSAT data for evaluation. However, they are included in the blended TROPOMI+GOSAT data available for download.

Our correction is built on top of the TROPOMI SRON v19 data that has already been bias-corrected with respect to SWIR surface albedo (Lorente et al., 2021). We compare these corrections in Appendix C.


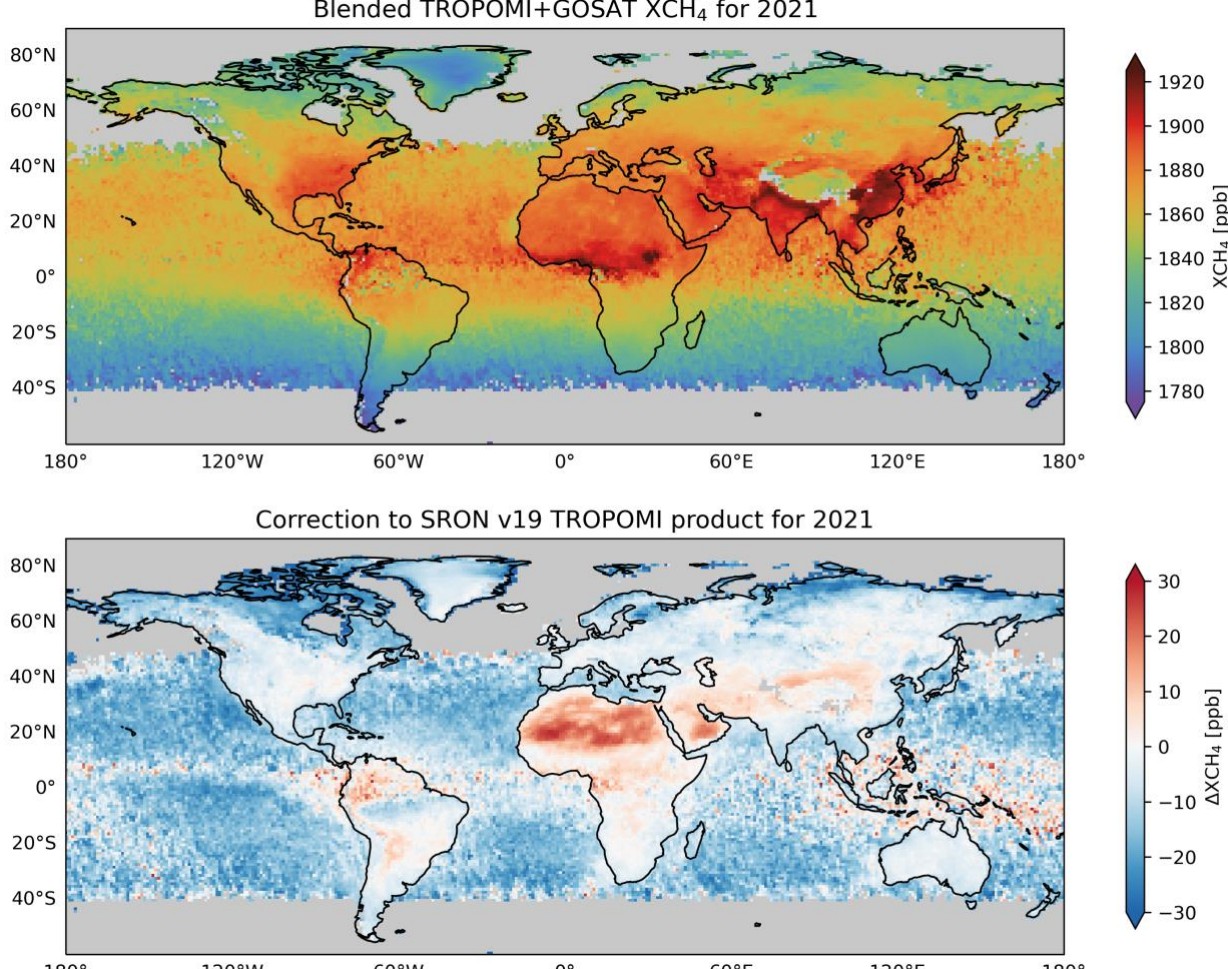

**Figure 8: Blended TROPOMI+GOSAT product for 2021 (top) and correction to the TROPOMI SRON v19 product (bottom). The data are annual means and plotted on a 1° × 1° grid with data south of 60°S cropped for visualization purposes. Grey areas have no TROPOMI data. The color bar in the bottom panel saturates at ± 30 ppb but there are outliers ranging from -57.8 ppb (west of Alaska) to +45.3 ppb (over the equatorial western Pacific).**

Figure 9 shows the seasonal variation of the correction for 2021. Upward correction over arid surfaces due to SWIR surface albedo is consistent across seasons, but there is still seasonality in the correction over these regions driven by dust emission (and thus the aerosol size parameter). As a result, the correction over North Africa is largest in spring-summer, and the correction over East Asian deserts is largest in late winter, reflecting the seasonality of dust emission (Shao and Dong, 2006; Senghor et al., 2017). There is large seasonal variation in the correction at high northern latitudes because of the low SWIR albedo of snow and ice-covered surfaces.



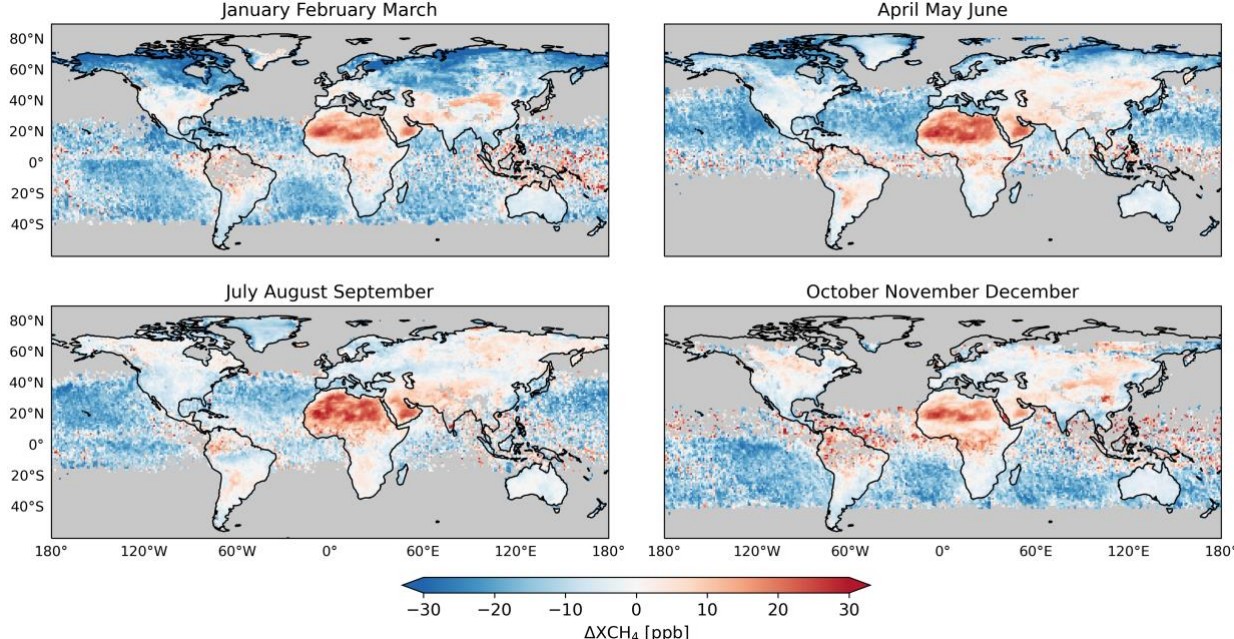

**Figure 9: Seasonal correction to the TROPOMI data in the blended TROPOMI+GOSAT product. The Figure shows the differences with the TROPOMI SRON v19 product averaged for each season in 2021. Data are plotted on a 1° × 1° grid and data south of 60°S are cropped for visualization purposes.**

Figure 10 illustrates the correction over the Arabian Peninsula with annual mean oversampled data on a 0.01° × 0.01° grid. The TROPOMI SRON v19 $XCH_4$ data show patterns that correlate with SWIR surface albedo, such as the $XCH_4$ gradients across Sudan and Saudi Arabia. These are removed in the blended product. The original data also show a number of hotspots over Iraq and Saudi Arabia that are removed in the blended product. These artifact enhancements tend to be related to the TROPOMI aerosol size distribution parameter, but they are more persistent than would be expected from aerosol plumes, suggesting that surface features might be aliasing into the aerosol retrieval. Other hotspots, such as those over Iran, are intensified in the blended product due to the high albedo in the region.

The TROPOMI SRON v19 data show some coastal artifacts that are apparent in Figure 10 and that are not always fully corrected in the blended TROPOMI+GOSAT product. Coastal scenes are difficult to retrieve in full-physics algorithms because of the subpixel albedo contrast between dark water and bright land. Despite our correction, Figure 10 shows that some coastal areas have persistent biases, most evidently along the southern coast of the Gulf of Aden. The ability of the ML model to correct coastal biases may be limited by the diversity of coastal conditions and the small number of TROPOMI and GOSAT coastal data pairs available for training. Data users can choose to mitigate coastal bias by filtering out a subset of TROPOMI scenes that contain both land and water (Appendix D).

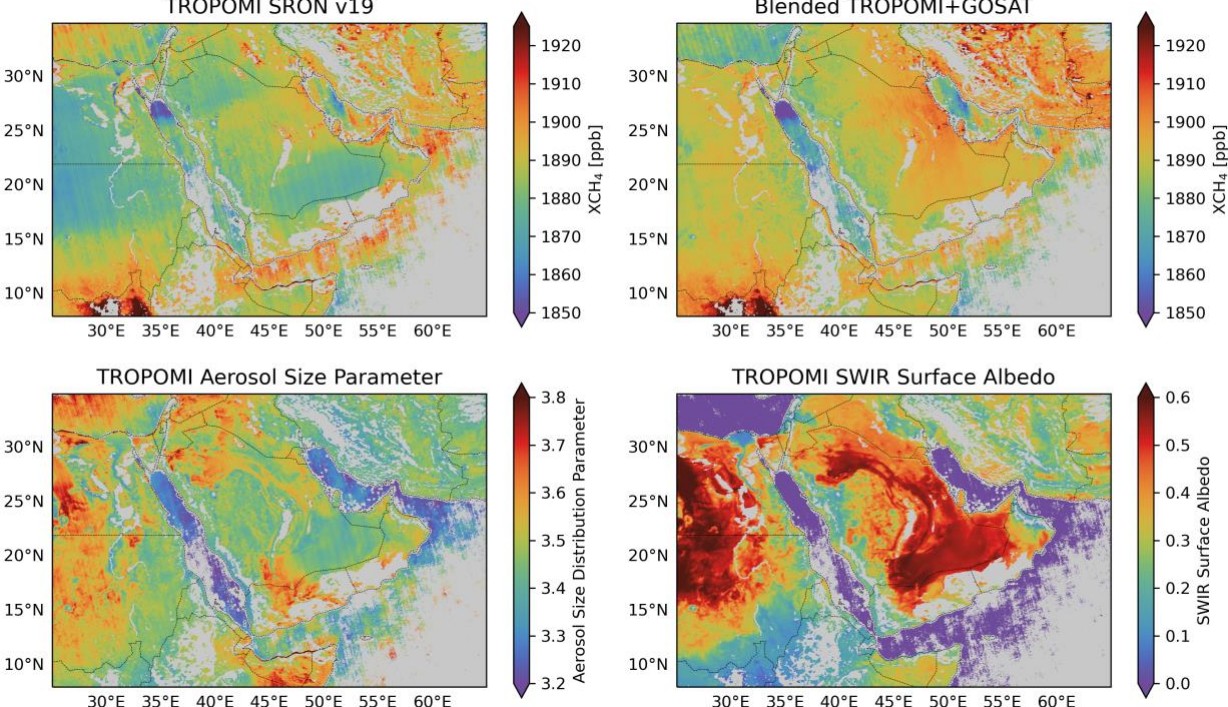

**Figure 10: Blended TROPOMI+GOSAT and TROPOMI SRON v19 data oversampled to a 0.01° × 0.01° grid over the Arabian Peninsula. Values are 2021 annual means. Also shown are the aerosol size distribution parameter and the SWIR surface albedo from the TROPOMI SRON v19 retrieval. Oversampling to increase spatial resolution was done with the tessellation method following Zhu et al. (2017). Grey areas have less than 10 individual satellite observations contributing to the average.**

Our blended product also corrects transient biases from striping and atmospheric scattering that may affect observations of hotspots and plumes. This is illustrated in Figure 11 with a single-orbit scene over Algeria on 15 December 2019 featuring a plume from an oil/gas ultra-emitting facility previously identified in the TROPOMI data by Lauvaux et al. (2022). There is strong striping along the orbit track in the original single-orbit data from TROPOMI (Liu et al., 2021; Schneising et al., 2023) and this is substantially reduced in our product. The plume, shown in the center of Figure 11, was partially overlain by cirrus clouds (observed by VIIRS) that were not filtered out in the TROPOMI retrieval but caused a low bias in the retrieved $XCH_4$. Our blended product corrects this cloud bias, enabling a better characterization of the plume to infer the source rate.

**Figure 11: Methane ultra-emitter plume detection in single-orbit TROPOMI data. The Figure shows a scene over Algeria sampled by orbit number 11252 on 15 December 2019. Missing data are shown in grey. The enhancement in the center of the image was identified by Lauvaux et al. (2022) from TROPOMI data as a plume from an ultra-emitting oil/gas facility with an approximate location at the white circle marker. The TROPOMI SRON v19 retrieval (top left panel) shows extensive striping and low values surrounding the plume that are biases from cirrus cloud reflectance (bottom left panel). This is effectively corrected in the blended TROPOMI+GOSAT product (right panels).**

## 5 Conclusions

We presented a new blended TROPOMI+GOSAT methane product that corrects spatially variable biases and artifacts in the TROPOMI satellite instrument observations of atmospheric methane (XCH$_4$) by referencing GOSAT observations. Our



blended product improves the reliability of inversions of TROPOMI data to infer methane emissions and identify methane super-emitters in single-orbit and time-averaged observations. It includes the full set of TROPOMI retrievals from January 2018 to present and is available for download (see Data Availability section).

The blended product was generated by training a machine learning (ML) model (LightGBM) to predict the difference
$\Delta$(TROPOMI-GOSAT) between co-located GOSAT and TROPOMI methane retrievals for 2018-2020, using TROPOMI retrieval parameters as the sole predictor variables. This enabled subsequent application of the ML model to compute $\Delta$(TROPOMI-GOSAT) for the full ensemble of TROPOMI data. The most important predictors of $\Delta$(TROPOMI-GOSAT) were found to be aerosol size, SWIR surface albedo, and across-track pixel index. The corrections were largest for observations with high albedos and coarse particles. Systematic downward correction averaging 12.7 ppb was found over water where the
GOSAT XCH$_4$ glint retrievals are lower than TROPOMI.

Evaluation with independent ground-based TCCON XCH$_4$ data shows that our blended TROPOMI+GOSAT product reduces the global mean bias in the TROPOMI data from 4.6 to -3.3 ppb, the variable bias from 6.0 to 5.2 ppb, and the single-retrieval precision from 13.8 to 11.7 ppb. However, the TCCON data are spatially limited and in particular do not sample
regions with SWIR surface albedos larger than 0.4 where the largest TROPOMI biases relative to GOSAT are found. Nevertheless, evaluation with the TCCON data confirms that TROPOMI retrieval biases are related to SWIR surface albedo and aerosol size parameter, and that the blended TROPOMI+GOSAT product successfully reduces these biases. Global evaluation of the blended TROPOMI+GOSAT product relative to GOSAT for 2021 as an independent test dataset shows a reduction in variable bias over land from 14.0 to 10.7 ppb on a 0.25° × 0.3125° grid (as might be used for regional inversions)
and from 12.8 to 9.6 ppb on a 2° × 2.5° grid (as might be used for global inversions).

Annual mean corrections in the blended product relative to the TROPOMI SRON v19 data exceed 10 ppb over the oceans, desert regions (notably North Africa), persistently cloudy regions (notably tropical wetlands), and seasonally snow-covered regions (notably high northern latitudes). Large-scale corrections are mostly driven by SWIR surface albedo. Fine-scale
inspection of the Arabian Peninsula reveals a number of annual mean hotspots in the original TROPOMI data that are removed in the blended product as artifacts. Some coastal artifacts remain in the blended product that can be filtered out at the discretion of the user.

The blended product also increases the quality of the single-orbit TROPOMI data by reducing striping and removing
transient biases from aerosol plumes and cirrus clouds. This can increase confidence in the identification of ultra-emitters from TROPOMI hotspots and the quantitative interpretation of plume observations to infer point source rates.

Our model is trained on data that is consistent with v02.04.00 of the operational TROPOMI methane product. The full TROPOMI observational record is being reprocessed to v02.04.00 as of this writing. Our correction will be applicable to all

past and future operational TROPOMI v02.04.00 data. Users can download our blended TROPOMI+GOSAT product for 2018-present (see Data Availability) or apply the correction themselves to the operational product. A new version of the TROPOMI retrieval would require retraining the ML algorithm.

The ML framework presented here can be extended to any pair of satellite instruments in which one instrument provides

a dense dataset and the other provides a more accurate but sparser dataset for the same variable. This situation often arises with a new satellite launch, as retrievals take time to mature, and an older, more established instrument may have been previously validated. Application of our approach to identify biases with the new instrument provides a far more spatially extensive evaluation than the traditional approach using surface sites or aircraft profiles. It also enables the identification of the critical retrieval parameters that should be improved in the new instrument. Finally, it generates a blended product that corrects data

from the new instrument.

## Appendix A: Adjustment of TROPOMI, GOSAT, and TCCON data to common averaging kernel sensitivities and prior vertical profiles

Unbiased intercomparison of XCH$_4$ values retrieved from TROPOMI, GOSAT, and TCCON requires adjustments for the different averaging kernel vertical sensitivities and prior vertical CH$_4$ concentration profiles used in the retrievals and reported

as part of the retrieval products. We follow Schneising et al. (2019) and Buchwitz et al. (2022) to make these adjustments.

The following notation will be used. Column-averaged dry air mixing ratios of methane (XCH$_4$) are denoted as $c$ with units of ppb. Vertical profiles of CH$_4$ sub column mixing ratios for pressure levels indexed by $l$ are denoted as $x^l$ and are either retrieved ($x_r^l$) or prior estimates ($x_a^l$) with units of ppb. Vertical profiles of averaging kernels describing sensitivity are denoted

by $A^l$ and are dimensionless. Pressure weights that map vertical profiles $x^l$ to XCH$_4$ are denoted as $h$ and are dimensionless. Subscripts $G$, $T$, and $F$ denote GOSAT, TROPOMI, and TCCON, respectively. $\Delta$ denotes the XCH$_4$ differences between pairs of instruments after adjustment to the same vertical sensitivities and prior estimates.

The GOSAT retrieval has more vertical pressure levels (19 or 20 depending on the retrieval) than TROPOMI (12, denoted

pressure layers). It is therefore better to interpolate $x^l$ from GOSAT to TROPOMI, following the principle of using the coarser vertical grid when comparing two different satellite retrievals (Rodgers and Connor, 2003). To calculate $\Delta$(TROPOMI-GOSAT), we first calculate what value $c_T^*$ TROPOMI would have retrieved with GOSAT's prior profile.





$$c_T^* = c_{T,r} + \sum_l h_T^l (1 - A_T^l)(x_{G,a}^l - x_{T,a}^l) \quad \text{(A1)}$$

Next, we calculate what value $c_G^*$ GOSAT would have retrieved with TROPOMI's vertical sensitivity.

$$c_G^* = \sum_l h_T^l \left( x_{G,a}^l + (x_{G,r}^l - x_{G,a}^l) A_T^l \right) \quad \text{(A2)}$$

Because the retrieved vertical profile of $CH_4$ is not reported for GOSAT, we estimate it here by scaling the prior profile by the ratio of retrieved to prior $XCH_4$ values.

$$x_{G,r} = x_{G,a} \frac{c_{G,r}}{c_{G,a}} \quad \text{(A3)}$$

Equations (A1) and (A2) require GOSAT's prior profile to be on the same pressure grid as TROPOMI. Interpolation is conducted from the 19 or 20 GOSAT pressure levels to the 12 TROPOMI pressure layers for this purpose. Equations (A1), (A2), and (A3) are then used to calculate Δ(TROPOMI-GOSAT).

$$\Delta(\text{TROPOMI} - \text{GOSAT}) = c_T^* - c_G^* \quad \text{(A4)}$$


The same procedure is used to calculate Δ(GOSAT-TCCON) and Δ(TROPOMI-TCCON). TCCON uses 51 pressure levels for its retrieval. For Δ(GOSAT-TCCON), we use TCCON's prior profile and GOSAT's averaging kernel sensitivities. For Δ(TROPOMI-TCCON), we use TCCON's prior profile and TROPOMI's averaging kernel sensitivities. An equation analogous to equation (A3) is used to estimate the retrieved TCCON $CH_4$ profile. Equations (A5) and (A6) thus calculate

Δ(GOSAT-TCCON) and Δ(TROPOMI-TCCON).

$$\Delta(\text{GOSAT} - \text{TCCON}) = \left[ c_{G,r} + \sum_l h_G^l (1 - A_G^l)(x_{F,a}^l - x_{G,a}^l) \right] - \left[ \sum_l h_G^l \left( x_{F,a}^l + (x_{F,r}^l - x_{F,a}^l) A_G^l \right) \right] \quad \text{(A5)}$$

$$\Delta(\text{TROPOMI} - \text{TCCON}) = \left[ c_{T,r} + \sum_l h_T^l (1 - A_T^l)(x_{F,a}^l - x_{T,a}^l) \right] - \left[ \sum_l h_T^l \left( x_{F,a}^l + (x_{F,r}^l - x_{F,a}^l) A_T^l \right) \right] \quad \text{(A6)}$$



## Appendix B: Evaluation with TCCON data

We evaluated the GOSAT, TROPOMI, and blended TROPOMI+GOSAT products with the independent TCCON data for 2018-2021, correcting for retrieval differences in prior information and vertical sensitivities (Appendix A). We use the TCCON GGG2020 data version (https://tccondata.org, last accessed 5 December 2022) and consider all 24 stations that have reported
measurements covering our study period of 2018-2021 (Table B1 and Figure 6).

**Table B1.** TCCON stations used for evaluation of the satellite data.

| Site (lat., lon.) | Elevation (km a.s.l.) | Data Reference |
| --- | --- | --- |
| Bremen (53.10, 8.85) | 0.03 | Notholt et al. (2022) |
| Burgos (18.53, 120.65) | 0.04 | Morino et al. (2022c) |
| East Trout Lake (54.35, -104.99) | 0.50 | Wunch et al. (2022) |
| Edwards (34.96, -117.88) | 0.70 | Iraci et al. (2022) |
| Eureka (80.05, -86.42) | 0.61 | Strong et al. (2022) |
| Garmisch (47.48, 11.06) | 0.74 | Sussmann and Rettinger (2023) |
| Hefei (31.90, 117.17) | 0.03 | Liu et al. (2022) |
| Izaña (28.30, -16.50) [a, b] | 2.37 | García et al. (2022) |
| JPL (34.20, -118.18) | 0.39 | Wennberg et al. (2022a) |
| Karlsruhe (49.10, 8.44) | 0.12 | Hase et al. (2022) |
| Lamont (36.60, -97.49) | 0.32 | Wennberg et al. (2022d) |
| Lauder (-45.04, 169.68) | 0.37 | Sherlock et al. (2022); Pollard et al. (2022) |
| Nicosia (35.14, 33.38) | 0.19 | Petri et al. (2023) |
| Ny-Ålesund (78.92, 11.92) [a, b] | 0.02 | Buschmann et al. (2022) |
| Orléans (47.97, 2.11) | 0.13 | Warneke et al. (2022) |
| Paris (48.85, 2.36) | 0.06 | Te et al. (2022) |
| Park Falls (45.95, -90.27) | 0.44 | Wennberg et al. (2022b) |
| Pasadena (34.14, -118.13) | 0.23 | Wennberg et al. (2022c) |
| Réunion Island (-20.90, 55.49) [b] | 0.09 | De Mazière et al. (2022) |
| Rikubetsu (43.46, 143.77) | 0.38 | Morino et al. (2022a) |
| Saga (33.24, 130.29) | 0.01 | Shiomi et al. (2022) |
| Sodankylä (67.37, 26.63) | 0.19 | Kivi et al. (2022) |
| Tsukuba (36.05, 140.12) | 0.03 | Morino et al. (2022b) |
| Xianghe (39.75, 116.96) | 0.04 | Yang et al. (2020); Zhou et al. (2022) |





<sup>a</sup> Excluded from the GOSAT evaluation due to a low number of pairs for comparison.

<sup>b</sup> Excluded from the TROPOMI and blended TROPOMI+GOSAT evaluations due to a low number of pairs for comparison.

The general evaluation framework is to identify co-located satellite and TCCON XCH$_4$ retrievals and compare these pairs. When evaluating TROPOMI or the blended TROPOMI+GOSAT product, satellite and TCCON pairs are defined to be those within 1 hour and 100 km of each other and a surface elevation difference of no more than 250 m (some of the TCCON stations

are on mountaintops). When evaluating the GOSAT product, satellite and TCCON pairs are defined to be those within 2 hours and 500 km of each other and a surface elevation difference of no more than 250 m. For all comparisons, a reduced radius of 50 km is used for the Edwards station (Schneising et al., 2019). We find 576975 TROPOMI-TCCON data pairs and 33071 GOSAT-TCCON data pairs. The TROPOMI-TCCON data pairs are also used to evaluate the blended TROPOMI+GOSAT product.


For each station, we take the mean and standard deviation of all values of Δ(satellite-TCCON) to yield a station bias and station precision. The mean bias is the average of the station biases. The variable bias is the standard deviation of the station biases. The retrieval precision is the average of the station precisions. These metrics are calculated for 2018-2021 and for all TCCON stations listed in Table B1 and mapped in Figure 6. Using a threshold of 100 satellite and TCCON pairs for a station

to be used, Izaña, Ny-Ålesund, and Réunion Island are excluded from the TROPOMI and blended TROPOMI+GOSAT analyses, while Izaña and Ny-Ålesund are excluded from the GOSAT analysis. The station biases and precisions by station are shown in Table B2.

**Table B2.** Comparison of satellite products with XCH$_4$ measured at TCCON stations. <sup>a</sup>

| | GOSAT | | TROPOMI | | Blended | |
|---|---|---|---|---|---|---|
| Site | μ (ppb) | σ (ppb) | μ (ppb) | σ (ppb) | μ (ppb) | σ (ppb) |
| Bremen | 0.6 | 14.3 | 5.5 | 13.0 | -3.3 | 9.9 |
| Burgos | -1.1 | 10.8 | 4.1 | 14.2 | -9.4 | 9.6 |
| East Trout Lake | 2.5 | 18.9 | 12.3 | 21.5 | 2.7 | 14.7 |
| Edwards | -1.1 | 10.6 | -2.9 | 9.9 | -7.5 | 8.0 |
| Eureka | 14.4 | 20.1 | 21.0 | 14.5 | 6.1 | 13.7 |
| Garmisch | 1.4 | 15.9 | 11.7 | 14.2 | 5.6 | 13.2 |
| Hefei | 2.0 | 20.5 | 7.0 | 13.4 | -0.3 | 12.0 |
| Izaña | -- | -- | -- | -- | -- | -- |
| JPL | -8.4 | 17.8 | -5.0 | 11.9 | -14.0 | 10.9 |



| | | | | | | |
|---|---|---|---|---|---|---|
| Karlsruhe | -3.3 | 14.9 | 3.0 | 12.7 | -3.7 | 10.2 |
| Lamont | -0.3 | 13.9 | -0.8 | 11.9 | -3.8 | 10.5 |
| Lauder | -1.5 | 9.7 | -2.6 | 13.8 | -9.5 | 10.6 |
| Nicosia | -0.3 | 12.6 | 4.1 | 13.0 | -4.4 | 12.0 |
| Ny-Ålesund | -- | -- | -- | -- | -- | -- |
| Orléans | -1.1 | 12.8 | 4.9 | 12.4 | -2.2 | 10.2 |
| Paris | -2.6 | 13.7 | 3.8 | 12.9 | -2.4 | 11.1 |
| Park Falls | 3.7 | 15.3 | 2.3 | 16.5 | -2.9 | 13.9 |
| Pasadena | -7.1 | 15.4 | -2.1 | 12.6 | -12.1 | 11.0 |
| Réunion Island | -7.3 | 11.0 | -- | -- | -- | -- |
| Rikubetsu | 10.4 | 17.0 | 7.2 | 14.1 | 2.8 | 13.4 |
| Saga | -2.1 | 14.3 | 13.0 | 12.4 | 1.8 | 11.3 |
| Sodankylä | 2.0 | 16.7 | 5.2 | 18.9 | -3.0 | 13.6 |
| Tsukuba | -5.0 | 12.9 | 1.2 | 11.5 | -5.1 | 10.6 |
| Xianghe | 3.4 | 20.4 | 3.7 | 15.3 | -4.6 | 15.6 |


[a] Mean ($\mu$) and standard deviation ($\sigma$) of the satellite-TCCON difference in $XCH_4$ for co-located data over the 2018-2021 period. Station locations are listed in Table B1 and shown in Figure 6. Satellite and TCCON data have been corrected to the same prior estimates and averaging kernel sensitivities (Appendix A). Dashes indicate insufficient co-located data (see text).

**Appendix C: Comparison to SRON Bias Correction**

As described in Lorente et al. (2021), the TROPOMI data from SRON v19 include a bias correction for SWIR surface albedo. In our study, we have used these data as our starting point for TROPOMI because they are presently being applied to produce the new operational product v02.04.00 covering the full TROPOMI record. This results in building a bias-correction (our work of $\Delta$(TROPOMI-GOSAT)) on top of another bias-correction (SRON's albedo correction).

The SRON SWIR surface albedo correction is derived using the "small area approximation," in which a few regions are selected around the globe where variation in SWIR surface albedo is observed but variation in $XCH_4$ is not expected. Referencing Aben et al. (2007), an albedo of 0.2 is selected as the best conditions for the retrieval and the correction is derived so that all retrievals in these regions match the retrieval at a SWIR surface albedo of 0.2.

Figure C1 shows the TROPOMI data before the SRON albedo bias correction as well as the magnitude of this correction. The sum of the two figures gives the TROPOMI SRON v19 data used in this study. Comparing the SRON albedo bias correction to Figure 6 suggests that it is pushing values of $XCH_4$ to be too low over bright surface (North Africa, Arabian Peninsula) and to be too high over dark surfaces (snow-covered scenes).

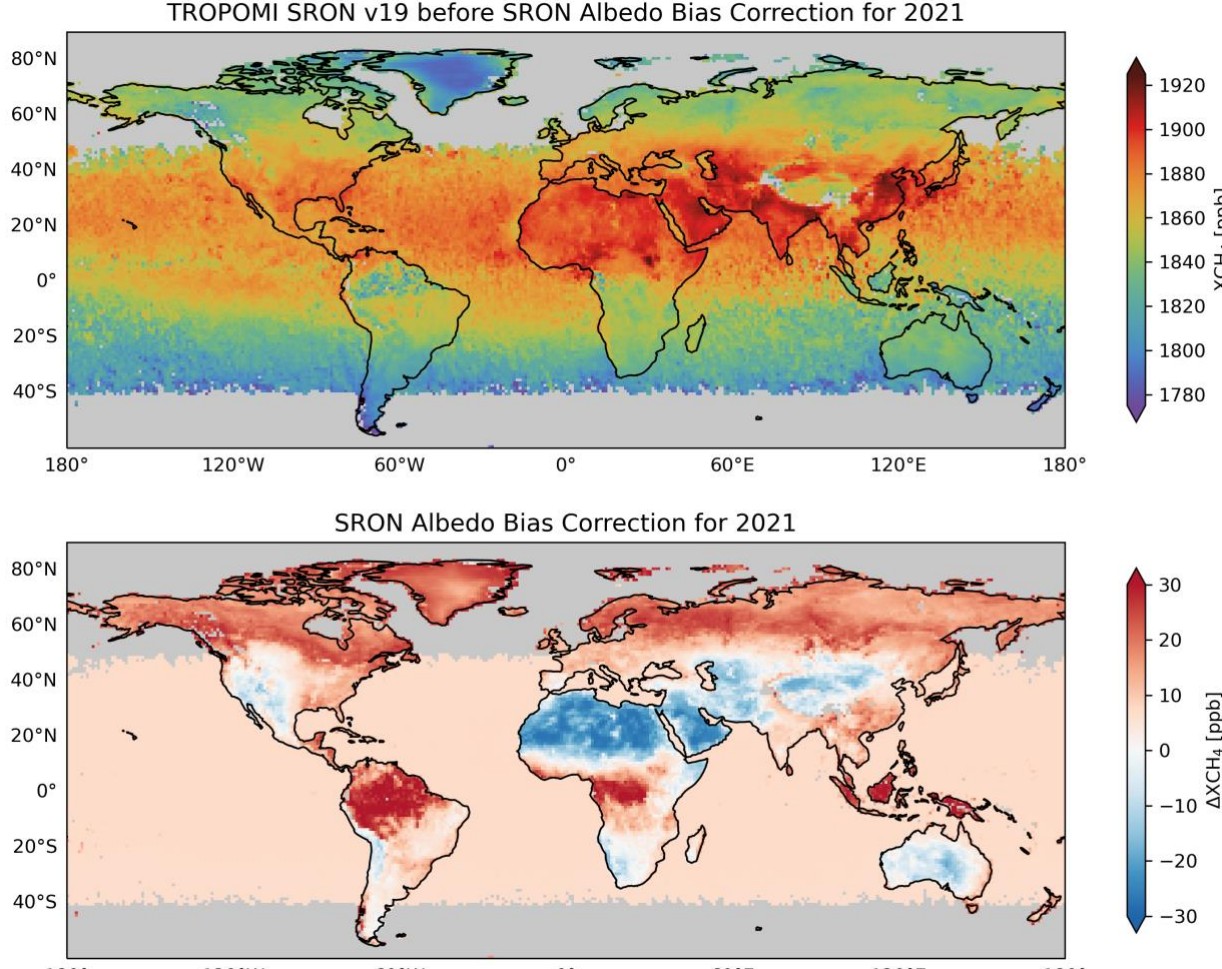

**Figure C1. TROPOMI SRON v19 XCH$_4$ product for 2021 before the SRON albedo bias correction (top) and the SRON albedo bias correction (bottom). The data are annual means and plotted on a 1° × 1° grid with data south of 60°S cropped for visualization purposes. Grey areas have no TROPOMI data. In the SRON files, the top plot corresponds to "xch4" and the sum of the top and bottom plot corresponds to "xch4_corrected."**

## Appendix D: Filtering Coastal TROPOMI Scenes

Retrieval pixels that include both land and water are problematic because of the subpixel albedo contrast. This can result in coastal biases, including for lakes and large rivers, that are not always successfully removed in our blended TROPOMI+GOSAT product. For example, in the top panel of Figure D1, there are enhancements of XCH$_4$ that outline the coast of North Africa. This can be fully avoided by filtering out all retrievals with a "landflag" value of 3 (pixel contains mostly



water with some land) and retrievals with a "landflag" value of 2 (pixel contains mostly land with some water) but this excludes 9% of the global data. Filtering out all pixels with "landflag" value of 3 (0.3% of data) and the subset of retrievals with "landflag" value of 2 and SWIR Chi-Squared greater than 20 (see Table 2) largely corrects coastal artifacts while excluding only 1% of the data. The middle panel of Figure D1 shows the pixels that are removed by this filter, and the bottom panel shows the blended product after removal of these pixels. We keep these coastal pixels in our blended product and leave it to

the user to decide what filters to apply.

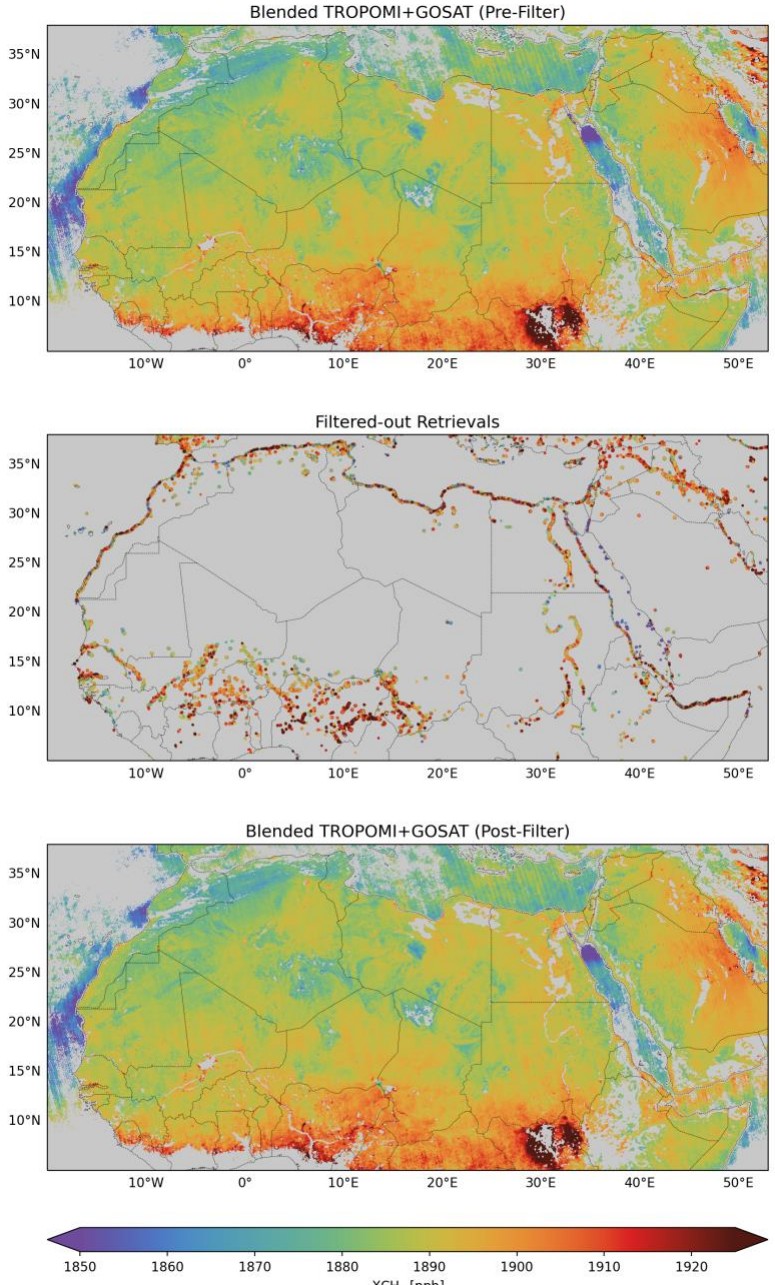

**Figure D1: Filtering of coastal pixels in the blended TROPOMI+GOSAT XCH₄ product. The top panel shows the unfiltered data over North Africa, oversampled on a 0.01° × 0.01° grid and averaged for 2021. The middle panel shows our coastal filter excluding all pixels with a "landflag" value of 3 and the subset of pixels with a "landflag" value of 2 and a SWIR Chi-Squared greater than 20 (Table 2). The bottom panel shows the filtered data. Grid cells with less than 10 individual observations contributing to the oversampled average are shown in grey in the top and bottom panels.**






*Code availability.* The code used for all portions of this project is available at

https://github.com/nicholasbalasus/blended_tropomi_gosat_methane and the preprint-version of this code is archived on Zenodo at https://zenodo.org/badge/latestdoi/587324684.

*Data availability.* The blended TROPOMI+GOSAT methane product has been generated to mimic the format of the SRON v19 product. The data from January 2018 to present will be available on Harvard Dataverse upon publication. The machine

learning model for correction will also be available on Harvard Dataverse upon publication. The TROPOMI data used here are available at https://ftp.sron.nl/open-access-data-2/TROPOMI/tropomi/ch4/19_446/ for 2018 to 2021. The GOSAT data used here are available at http://dx.doi.org/10.5285/18ef8247f52a4cb6a14013f8235cc1eb for 2009 to 2021. The TCCON data were obtained from the TCCON Data Archive hosted by CaltechDATA at https://tccondata.org.

*Author contributions.* NB and DJJ designed the study. NB performed the analysis with contributions from AL, JDM, RJP, HB, ZC, MMK, HN, and DJV. NB and DJJ led the writing of the paper with contributions from all co-authors.

*Acknowledgements.* We gratefully acknowledge TCCON site PIs for the data used in this work and their guidance on its use. This research used the ALICE high-performance computing facility at the University of Leicester for the GOSAT retrievals

and analysis. We thank the Japanese Aerospace Exploration Agency, National Institute for Environmental Studies and the Ministry of Environment for the GOSAT data and their continuous support as part of the Joint Research Agreement.

*Competing interests.* The authors declare that they have no conflict of interest.

*Financial support.* This work was supported by the NASA Carbon Monitoring System. Nicholas Balasus was supported by the Department of Defense (DoD) through the National Defense Science & Engineering Graduate (NDSEG) Fellowship Program. Robert J. Parker and Hartmut Boesch are funded via the UK National Centre for Earth Observation (grant nos. NE/R016518/1 and NE/N018079/1), as well as from the ESA GHG-CCI and Copernicus C3S projects (grant no. C3S2_312a_Lot2).

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
