# Peer review of "A blended TROPOMI+GOSAT satellite data product for atmospheric methane using machine learning to correct retrieval biases"

_Atmospheric Measurement Techniques, 2023_

## Author Response (AR1)

**Applicable to Both Referees)**

We thank the reviewers for their feedback on the manuscript. We have included responses to the reviewer comments below. Reviewer comments are in *italics* and our responses are in **bold** (with text revisions **underlined**).

Since the time of the submission, the reprocessing of the operational TROPOMI record has been completed. As such, we have changed our underlying TROPOMI data from SRON v19 to operational v02.04.00 (allowing us to extend the data we distribute to the present). The processors are nearly identical (the SRON v19 research product served as a beta form of the operational v02.04.00 product), though the TROPOMI v02.04.00 record begins in April 2018 instead of January 2018. As a result of this change, some of the metrics and figures have changed slightly, but none of the conclusion are altered.

**Anonymous Referee #1)**
*This paper describes a method to empirically correct TROPOMI XCH4 biases using collocated GOSAT XCH4 estimates as a reference. The method is based on a Machine learning technique and uses a number of variables from TROPOMI as predictors.*

*The paper is very well written. The objectives a clear, the method is well described with the proper level of details, and the conclusions follow from the data analysis. The method and result can be of interest to the scientific community that develops remote sensing products for the monitoring of greenhouse gases in the atmosphere.*

**We thank the reviewer for their comments.**

*There are a large number of figure, some of which are not necessary for the demonstration. I suggest that the following figures can be removed of put in a supplementary material*

*Figure 4*

*Figure 5 (top) as it is redundant with Figure 1*

*Figure 6*

*Figure 9*

**We thank the reviewer for the suggestion but we think that these Figures are important and that creating an SI or deleting them would weaken the paper. We note that there is a difference between Figure 1 (2018-2021 data, used to introduce the problem) and the top of Figure 5 (2021 data, used to show the test data).**

**Anonymous Referee #2)**
*GENERAL COMMENTS*

*A single instrument onboard a spacecraft can provide uniform quality data globally, but its spatial coverage and frequency are stll limited. As remote sensing data has larger uncertainties than in-situ measurements, present number of the satellite data is not enough for analysis. The blended product approach using multiple sensors described in this paper is a key technique to solve the above issue.*

**We thank the reviewer for their comments.**

*It will be helpful to summarize why TROPOMI have larger spatially variable biases. Is it due to lower spectral resolution or more complicated spectral response? Can proxy method remove the error due to light path modifications? If the latter is true, is there significant difference between results from full physics and proxy methods?*

*The description why GOSAT data can be used as reference will also help. GOSAT has following advantages for the CH4 retrieval.*

1. *Higher spectral resolution*

2. *Fourier transform spectrometers provide simpler line-shape function than imaging grating spectrometers.*

3. *Mechanical cross track pointing can provide uniform spectral response over +/- 35 deg at the expense of data density.*

4. *Long term record since 2009*

**The spatially variable biases in TROPOMI relative to GOSAT come from a variety of factors related to radiative transfer, including the different spectral resolutions, types of spectrometer, and retrieval method (as necessitated by the spectral viewing window). We attempt to diagnose the reasons for Δ(TROPOMI-GOSAT) with our feature importance analysis, though our methods have limitations for causal inference. We have modified the text to extend upon the benefits of GOSAT that the reviewer has pointed out.**

**"The TROPOMI instrument was launched in 2017 and provides global daily coverage, but it is more subject to biases than GOSAT because it uses a different spectral viewing window, has coarser spectral resolution, and relies on an array of detectors (Jacob et al., 2022)."**

**"GOSAT utilizes a Fourier transform spectrometer with mechanical cross-track pointing, providing a uniform spectral response for its observations and consistent high-quality data from 2009 to present (Kuze et al., 2016)."**

**"TROPOMI provides global daily coverage in continuous $5.5 \times 7$ km² (nadir) pixels, increasing the data density relative to GOSAT by more than two orders of magnitude through the use of an imaging grating spectrometer."**

*I recommend the publication after minor revisions.*

*SPECIFIC COMMENTS*

*(1) Page 9 Figure 2, "Across track pixel index" , "aerosol size distribution" and "Surface albedo"*

*Does the bias have across track angle dependency (geometry dependent) or specific pixels have biases (detector non-uniformity or bad pixels)?*

*Are there any systematic differences in biases between the east (backward reflection and scattering) and west looking (forward) scenes? Geometry of the sun, surface, and satellite affects scattering by aerosol and thin clouds. Over the Sahara Desert, both surface albedo and aerosol density high. Does the Machine Learning can distinguish from seasonal variation?*

The origin of the striping seen in the TROPOMI data is not fully understood and is not constant from orbit-to-orbit (Borsdorff et al., 2019), though Schneising et al. (2023) suggest detector non-uniformity to be the culprit. We use across-track pixel index to try to predict this bias. However, as the reviewer suggests, it also provides some important information on geometry (which can be redundant with information from the relative azimuth angle and solar zenith angle), leading to the high feature importance for across-track pixel index (Figure 2). This is part of the information that the machine learning can leverage to form a correction that varies seasonally (such as that over the Sahara Desert, see Figure 9). We have updated the text as follows to reflect this:

"The importance of the across-track pixel index reflects the striping patterns present in TROPOMI retrievals, which change from orbit to orbit (Borsdorff et al., 2018; 2019). Additionally, the across-track pixel index provides information about the viewing geometry of TROPOMI."

As to the comment on "bad pixels," these are filtered out in the level 2 processing.

*(2) Page 10, Line 236*

*"The RMSE is smaller than would be expected the precision .."*

*It is not clear for me. Do the authors mean "simply add "15.0 ppb of GOSAT and 13,8 ppb of TROPOMI" (Sqrt (15.0^2+13.8^2)?*

This is correct. We've added the numbers from Table 1 now to make this more clear. Note that the numbers are slightly different due to the change from SRON v19 to TROPOMI v02.04.00.

"The RMSE is smaller than would be expected from the precision of Δ(TROPOMI-GOSAT) derived by adding the precisions of the individual TROPOMI (14.5 ppb) and GOSAT (14.9 ppb) retrievals relative to TCCON in quadrature (20.8 ppb; Table 1)."

*(3) Page 20, Line 414 "single-orbit TROPOMI"*

*What do the authors mean by "single orbit"? Please describe the definition.*

We use the term "single orbit" to refer to data that has not been spatially- or temporally-averaged (as has been done for the other figures). We update the text for an earlier reference to "single orbit" to be more clear.

"This is illustrated in Figure 11 with a single-orbit scene (no temporal or spatial averaging) over Algeria on 15 December 2019 featuring a plume from an oil/gas ultra-emitting facility previously identified in the TROPOMI data by Lauvaux et al. (2022)."

*TECHNICAL CORRECTIONS*

*(1) Page 2, Line 49 "near-unit sensitivity down to the surface"*

*Does this mean "vertically-uniform sensitivity form top of the atmosphere down to the surface" or "from the troposphere down to the surface?"*

Thank you for pointing this out. We have updated the text to follow this suggestion.

"There are strong methane absorption features at 1.65 μm and 2.3 μm, enabling retrieval of the atmospheric methane column with near vertically-uniform sensitivity from the top of the atmosphere down to the surface under clear-sky conditions (Frankenberg et al., 2005)."

New References)

Borsdorff, T., aan de Brugh, J., Schneider, A., Lorente, A., Birk, M., Wagner, G., Kivi, R., Hase, F., Feist, D. G., Sussmann, R., Rettinger, M., Wunch, D., Warneke, T., and Landgraf, J.: Improving the TROPOMI CO data product: update of the spectroscopic database and destriping of single orbits, Atmos. Meas. Tech., 10, 5443-5455, https://doi.org/10.5194/amt-12-5443-2019, 2019.